# HIV-1 Accessory Proteins Impart a Modest Interferon Response and Upregulate Cell Cycle-Related Genes in Macrophages

**DOI:** 10.3390/pathogens11020163

**Published:** 2022-01-26

**Authors:** Laura J. Martins, Matthew A. Szaniawski, Elizabeth S. C. P. Williams, Mayte Coiras, Timothy M. Hanley, Vicente Planelles

**Affiliations:** 1Division of Microbiology and Immunology, Department of Pathology, University of Utah School of Medicine, Salt Lake City, UT 84112, USA; laura.martins@path.utah.edu (L.J.M.); elizabeth.williams@path.utah.edu (E.S.C.P.W.); 2Department of Pediatrics, University of Utah School of Medicine, Salt Lake City, UT 84112, USA; matt.szaniawski@hsc.utah.edu; 3AIDS Immunopathology Unit, National Center of Microbiology (CNM) Instituto de Salud Carlos III (ISDIII), 28222 Madrid, Spain; mcoiras@isciii.es; 4Division of Hematopathology, Department of Pathology, University of Utah School of Medicine, Salt Lake City, UT 84112, USA

**Keywords:** HIV-1, interferon, cell cycle, innate immune, transcriptome, accessory proteins

## Abstract

HIV-1 infection of myeloid cells is associated with the induction of an IFN response. How HIV-1 manipulates and subverts the IFN response is of key interest for the design of therapeutics to improve immune function and mitigate immune dysregulation in people living with HIV. HIV-1 accessory genes function to improve viral fitness by altering host pathways in ways that enable transmission to occur without interference from the immune response. We previously described changes in transcriptomes from HIV-1 infected and from IFN-stimulated macrophages and noted that transcription of IFN-regulated genes and genes related to cell cycle processes were upregulated during HIV-1 infection. In the present study, we sought to define the roles of individual viral accessory genes in upregulation of IFN-regulated and cell cycle-related genes using RNA sequencing. We observed that Vif induces a set of genes involved in mitotic processes and that these genes are potently downregulated upon stimulation with type-I and -II IFNs. Vpr also upregulated cell cycle-related genes and was largely responsible for inducing an attenuated IFN response. We note that the induced IFN response most closely resembled a type-III IFN response. Vpu and Nef-regulated smaller sets of genes whose transcriptomic signatures upon infection related to cytokine and chemokine processes. This work provides more insight regarding processes that are manipulated by HIV-1 accessory proteins at the transcriptional level.

## 1. Introduction

Human immunodeficiency type 1 (HIV-1) targets several key immune cells which respond to viral insult through the generation of an innate immune response [1]. This response, driven by detection of pathogen-associated molecular patterns (PAMPs), leads to the formation of an antiviral state [1]. In many HIV-1 target cells, a type-I interferon (IFN) response predominates, which is characterized by the expression of numerous IFN-regulated genes (IRGs) [2]. The resulting antiviral state is subsequently established in both infected and bystander cells [3]. While there has been a significant effort to understand the role of type-I IFNs in this process, less is understood about the landscape of additional IFN subtypes (i.e., II and III), their impact on HIV-1 infection, and whether the innate immune response in macrophages more closely resembles their IRG signatures [4,5,6].

Macrophages represent an important HIV-1 target cell type because of their ubiquitous presence throughout tissues and organ systems (including anatomic regions important for viral transmission), their role in acute and chronic phases of infection, and their contribution to viral persistence over time [7,8]. Indeed, macrophages are capable of sensing HIV-1 infection at several steps of the HIV-1 life cycle, and activation of signaling pathways consistent with type-I IFN sensing have been documented [2]. Interestingly, several studies have failed to detect type-I IFN production itself at the protein or mRNA level during in vitro infection of MDM with HIV-1, despite the induction of numerous IRGs [2,9].

IFNs exert clear inhibitory effects during the acute phase of viral infection. On the other hand, it has been speculated that IFNs in the chronic setting may contribute to reservoir generation, maintenance, and persistence [6,10,11,12]. Macrophages are of particular interest in this context as they are both highly responsive to IFN stimulation and presumed to be responsible for its production in the acute and chronic phases [9,13]. Type-I IFNs include a diverse set of signaling molecules which comprise numerous IFN-α subtypes, IFN-β, IFN-ε, IFN-κ, and IFN-ω [14,15]. IFN-γ is the sole member of the type-II IFN class, and type-III IFNs include four IFN-λ subtypes [14,15,16]. Engagement of all IFN types with their cognate receptors induces a variety of signaling events which collectively trigger JAK-STAT signaling cascades, ultimately leading to broad transcriptional changes and expression of a plethora of antiviral genes [17].

A unique aspect of HIV-1 is its complement of accessory genes, known individually as Vif, Vpu, Vpr, and Nef (and, additionally, Vpx in HIV-2) [18,19]. Collectively, these genes overcome replication inhibition imposed by host cell restriction factors including SAMHD1 (Vpx), APOBEC3G (Vif), and tetherin (Vpu) [18,19,20,21,22]. Several restriction factors have also been documented to reduce type-I IFN expression in HIV-1 target cells [23]. For example, Vif and Vpr have been demonstrated to interact directly with TBK1 to prevent its phosphorylation in myeloid cells and, therefore, limit the IFN response [23]. Whether these accessory proteins interfere with the generation of type-I or type-II IFN responses remains unclear and was a major aim of this study.

Here, we sought to further characterize the response of monocyte-derived macrophages (MDMs) after in vitro infection by HIV-1. We further aimed at dissecting the role of HIV-1 accessory proteins and their effects on the transcriptome induced during in vitro viral infection.

## 2. Results

### 2.1. Infection of Monocyte Derived Macrophages with HIV-1_Bal_ Induces an IFN-Like Response

We sought to better understand how changes in the macrophage transcriptome induced by HIV-1 infection compare with changes after stimulation with interferons. To that end, we performed RNA-seq analysis on MDMs infected with R5 tropic HIV-1 (HIV-1_Bal_) or treated with type-I (IFN-α or IFN-ε), type-II (IFN-γ), or type-III (IFN-λ), referred to as dataset #1 (Appendix A) [9,24]. In total, 78 genes were downregulated and 301 genes were upregulated upon infection of MDMs with HIV-1_Bal_. HIV-1 up- and downregulated genes were compared to genes downregulated upon stimulation with individual interferons (Figure 1a). HIV-1-upregulated genes were present among genes upregulated by all four IFNs tested, with IFN-λ stimulation sharing the largest number of common upregulated genes (Figure 1b). The transcriptome profile resulting from infection HIV-1_Bal_ infection of MDMs including both up- and downregulated genes, most closely resembles a type-III IFN response as observed when hierarchical clustering heat map analysis was performed (see Figure 1b). This is interesting given that type-III IFNs are much less potent at inhibiting HIV-I infection of macrophages that are type-I or -II IFNs [9]. We quantitated differences between transcriptome changes induced after stimulation with each interferon and corresponding changes induced upon HIV-1_Bal_ infection using the Euclidean method: $\sqrt{\overset{n}{\underset{i=1}{\sum }}{\left(q_{i}-p_{i}\right)}^{2}}$ (where *p* and *q* are gene expression levels induced by HIV-1 infection (*p*) or IFN stimulation (*q*) and *i* denotes the gene in the datasets that are being compared) (Figure 1c). These results indicate that, at 24 h post infection, HIV-1 induces a transcriptome response that includes signatures resembling type-I, -II, and -III interferon stimulation, with a type-III response being most similar to HIV-1. Not surprisingly, the most represented pathways enriched by HIV-1 infection of MDMs include type-I interferon signaling pathways and pathways related to other immune functions, such as chemokine signaling (Figure 1d and Appendix A). Transcription factors enriched among HIV-1 DEGs include forkhead box M1 (*FOXM1*), interferon regulatory factor 8 (*IRF8*), interferon regulatory factor 1 (*IRF1*), and early growth response 1 (*EGR1*; Figure 1e and Appendix A). *IRF1* and *IRF8* are transcription factors whose targets are associated with interferon responses. *EGR1* regulates genes in response to growth factors and DNA damage, including cytokines and hormones. *FOXM1* is a master regulator of the cell cycle and is essential for DNA replication, mitosis, and DNA damage repair [25,26]. Eukaryotic Translation Initiation Factor 2 Alpha Kinase 2 (*EIFAK2*), also called protein kinase RNA-activated *(**PKR*) is the most highly enriched kinase among genes differentially regulated by HIV-1_Bal_ infection of MDMs (Figure 1f and Appendix A). PRK is a kinase that regulates translation of HIV-1 transcripts and, when activated, has potent antiviral activity [27]. The RNA-seq dataset can be found in Appendix A).

### 2.2. HIV-1 Vif and Vpr Induce Significant Transcriptome Changes in Macrophages Infected with HIV-1

To assess the contribution of viral accessory proteins to the transcriptome changes observed upon HIV-1 infection we infected MDMs with a recombinant HIV-1_Bal_ clone encoding murine CD24 (mCD24), herein referred to as WT or with accessory gene mutant HIV-1 viruses (dataset #2). Three replicate samples of MDMs were mock infected, infected with WT virus, or infected with accessory gene mutant HIV-1_Bal_. Western blotting was used to verify the expression, or lack thereof, of individual accessory proteins (see Appendix A). MDMs infected with WT and mutant viruses expressed similar levels of cell surface mCD24 protein expression (Figure 2a) and mCD24 transcripts (Figure 2b) with between 40 and 60% of MDMs productively infected at 24 h post infection for all experiments. We performed principal component analysis (PCA) to determine bulk differences in expression profiles between MDMs infected with WT and mutant viruses compared to mock infection (Figure 2c). Transcriptomes from HIV-1_Bal_ΔVif, ΔVif (brown triangles) and HIV-1_Bal_ΔVpr, ΔVpr (green squares), were most different from WT infection (pink stars), whereas the transcriptome from HIV-1_Bal_ΔNef, ΔNef (orange circles) infected cells was the most similar to WT infection. We quantified the number of differentially expressed genes (DEGs) from comparisons between WT and accessory gene mutant viral infections (Figure 2d). Vpr and Vif accounted for the largest number of transcriptome changes with 285 and 191 DEGs, respectively. We then compared DEGs from each mutant to those present in dataset #1 (Appendix A [9,24] to assess the relevance of the IFN response to the transcriptional changes (Figure 2e). Vpr and Vif differentially regulated the largest number of IFN-regulated genes (IRGs) with 199 and 176 IRGs, respectively. These results suggest that accessory genes play a significant role in the regulation of IRGs upon HIV-1 infection of macrophages.

### 2.3. Validation of Differential Expression of IDO1 and PLK1

We sought to confirm differential expression of representative IRGs and cell cycle-related genes using real-time quantitative polymerase chain reaction (RT-qPCR). Indoleamine 2,3-dioxygenase 1 (*IDO1*) is an enzyme that catalyzes the rate limiting step in tryptophan catabolism and a broadly acting pathogen restriction factor that limits the availability of tryptophan to pathogens [28,29]. *IDO1* is highly induced in our RNA-seq data from MDM cultures that were either stimulated with type-I or -II interferons (not shown here) or infected with WT, but not ΔVpr, ΔVif, or ΔVpu (Figure 3a). We also selected Polo-Like Kinase 1 (*PLK1*) based on its known role in phosphorylating numerous proteins involved with mitotic cell signaling pathways. *PLK1* expression was suppressed in MDMs infected with WT virus and with each accessory gene HIV-1 mutant (Figure 3b). However, *PLK1* expression was potently suppressed in MDMs infected with ΔVif and only weakly suppressed in MDMs infected with ΔVpr in our RNA-seq data (Figure 3a). We analyzed three aliquots of RNA samples that were also subjected to RNA-seq for expression of *IDO1*, *PLK1,* and *GAPDH*, as a normalizing gene and confirm that *IDO1* is upregulated upon HIV-1 infection of MDMs in a manner dependent on *Vpr* and *Vif* (Figure 3b) and that *PLK1* expression is suppressed upon infection with WT virus and more potently suppressed upon infection with ΔVif (Figure 3c).

### 2.4. HIV-1 Vif Induces Transcriptome Changes from Genes Related to Mitosis and the Microtubule Network in Infected Macrophages

To better understand relationships between the 191 Vif-regulated genes and biological processes affected by Vif, we performed a protein network and pathway analysis. We analyzed those genes differentially expressed by < or > 0.6 log2FC with a significance of *p* < 0.05 when comparing transcriptomes from WT HIV-1_Bal_ infected MDMs to HIV-1_Bal_ ΔVif infected MDMs. Known interactions between the Vif-regulated DEGs were identified using a search tool for the retrieval of interacting genes (STRING) displayed as a protein interaction network (PIN) (Figure 4a). A significant number of interactions among these DEGs are the result of their contributions to cell cycle processes (purple hemisphere border) and a smaller subset are genes involved in immune responses (green hemisphere border). Specifically, Vif-regulated genes are highly enriched for pathways (Figure 4b and Appendix A), transcription factors (Figure 4c and Appendix A), and kinases (Figure 4d and Appendix A) related to mitosis. E2F transcription factor 4 (*E2F4*) and *FOXM1* were highly enriched among Vif-regulated genes. *E2F4* is a transcription factor that suppresses proliferation-associated genes and controls transcription of cell cycle-related genes. Polo-like kinase 1 (*PLK1*) and aurora kinase B (*AURKB)* are just 2 of 58 kinases identified as being significantly enriched among Vif-regulated DEGs. *PLK1* and *AURKB* regulate numerous cell-signaling molecules, transcription factors, and components of the complex machinery required to complete mitosis, such as kinesins and centromere proteins. This observation is unusual given that macrophages are terminally differentiated, non-cycling cells. However, the microtubule network is required for macrophage functions such as motility and phagocytosis and is necessary for HIV-1 capsid trafficking [30,31,32]. Vif-regulated genes also bear resemblance to genes differentially expressed in IFN-λ treated macrophages (Figure 4e,f). Interestingly, 50% or 91 of 180 Vif upregulated genes are genes that are potently downregulated by type-I and -II IFNs in dataset #1, <1.5 log2FC, p_adj_ < 0.05 compared to unstimulated MDMs. We further investigated this subset of 91 genes upregulated by Vif and demonstrated that Vif-upregulated genes within this subset are remarkably similar to those induced by IFN-λ (Figure 5a,b) and demonstrated that the IRG signature is dominated by pathways (Figure 5c), transcription factors (Figure 5d), and kinases (Figure 5e) related to the cell cycle.

Because we uncovered the unusual finding that mitosis related genes were upregulated by HIV-1 infection and IFN-λ stimulation of macrophages and also downregulated by type-I and -II IFNs, which are known to be produced during HIV-1 infection, we compared our findings to those from a similar study from another laboratory to determine the relevance of this finding. The most analogous study was reported by Deshiere et al. [33]. We compared four sets of DEGs from MDMs infected with WT HIV-1_Bal_ and sorted using mCD24 (also referred to as heat stable antigen (HSA): HSA+ 36hpi (Appendix A), HSA- 36 hpi (Appendix A), HSA+ 6 hpi (Appendix A), HSA- 6 hpi (Appendix A)), to IRGs from our dataset. HSA+ cells are productively infected and HSA- cells are cells exposed to HIV-1 but not productively infected. We found a similar pattern of altered gene regulation in productively infected cells at 36 hpi (HSA+ 36 hpi), but not in MDMs exposed to HIV-1 but not productively infected (HSA-) (Appendix A). In fact, the transcriptome from HSA- MDMs purified after 36 h of infection was virtually free of IRGs present in our dataset. This unique signature of genes with remarkably differing expression responses to type-III relative to type-I or type-II is also surprising given similarities in cell signaling resulting from type-I, -II, and -III IFN stimulation. To rule out the possibility that, within the context of HIV-1 infection of macrophages, the overall IRG signature is the result of a type-I or -II IFN induction that differs from concentrations that we tested, we analyzed expression changes of kinesin family member 20A (*KIF20A)*, an IRG that is potently upregulated by *vif* and potently downregulated upon stimulation with 25 ng/mL IFN-α, IFN-ε, or IFN-γ (Figure 5a), using multiple doses (Appendix A). *KIF20A* is a mitotic kinesin required for cytokinesis. Our data demonstrate that KIF20A expression is potently downregulated by IFN-α and IFN-γ and is modestly upregulated by IFN-λ within this broader concentration range. It is, therefore, unlikely that the observed IFN-λ-like IRG signature upon HIV-1 infection of MDMs is a result of induction of IFN-α or IFN-γ.

### 2.5. HIV-1Vpr Induces an Attenuated IFN Response and Regulates Host Cell Cycle Genes in MDMs

We then performed protein interaction mapping and pathway analysis on the 285 Vpr-regulated genes, which included those with a change < or > 0.06 log2FC with p_adj_ < 0.05 between the transcriptomes of WT HIV-1_Bal_ and HIV-1_Bal_ΔVpr infected MDMs. Protein interaction network mapping revealed clusters of genes that are involved in immune responses and cell cycle processes (Figure 6a). The 10 most highly enriched pathways included IFN signaling pathways, hormone signaling, cytokine processes, and DNA replication (Figure 6b and Appendix A). Identified transcription factors and kinases reflect changes in both interferon signaling and cell cycle-related pathways. *E2F4*, a transcription factor associated with proliferation was the most significantly enriched among both Vpr- and Vif-DEGs. Transcription factors RUNX family transcription factor 1 (*RUNX1*) and IFN regulatory factor 1 (*IRF1*) are enriched within the Vpr set of DEGs, with 28 and 7 target genes mapped to these transcription factors, respectively (Figure 6c and Appendix A). *RUNX1* and *IRF1* modulate genes related to immune responses, including IRGs such as triggering receptor expressed on myeloid cells 1 (*TREM1*) and interferon-stimulated gene 15 (*ISG15*). Additionally, receptor interacting serine/threonine kinase 2 (*RIPK2*) and interleukin 1 receptor associated kinase 2 (*IRAK2*) were present in the list of significantly enriched kinases among Vpr-regulated genes, but not among Vif-regulated genes (Figure 4d and Appendix A). *RIPK2* and *IRAK2* are receptor associated serine/threonine kinases that lead to activation of NF-kB upon stimulation of their cognate receptors. The heatmap comparison of Vpr-regulated genes with transcriptome changes upon treatment of MDMs with type-I, -II, and -III IFNs (Figure 6e) revealed the presence of a subset of 44 Vpr upregulated genes, which were potently downregulated in the presence of type-I or -II IFNs.

We further investigated this subset of 44 genes (Figure 7a) to determine which pathways were involved and if these pathways were similar to those observed for the Vif upregulated and IFN (type-I and/or -II) downregulated subset. Cell cycle pathways were strongly associated with this subset of Vpr upregulated genes (Figure 7c). The *E2F4* transcription factor was highly enriched within this subset as it was in the subset of Vif upregulated and IFN type-I and -II downregulated genes. However, unlike the Vif subset (Figure 5d), the *FOXM1* transcription factor was not significantly associated with the Vpr subset (Figure 7d). *E2F4* target genes comprised a majority of genes that are upregulated by both Vif and Vpr but potently suppressed by type-I and -II IFNs (Figure 7e). Interestingly, Vpr upregulated *IL1B,* a potent inflammatory cytokine that has been reported to be suppressed by type-I and -II IFNs. We compared transcriptomic profiles generated by infection of the accessory gene mutants (blue bars) to those generated by WT HIV infection (orange bars) and stimulation with IFN-λ (grey bars) in order to assess the contribution of each accessory gene to expression changes generated by infection with WT HIV-1_Bal_ compared to mock infection using overlay plots (Figure 8). Of the 73 genes with differential expression upon infection with WT, 68 were present in the list of genes differentially regulated upon stimulation with type-I, -II, and/or type-III IFNs from dataset #1. Deletion of *vpr* from the HIV-1 genome resulted in the largest transcriptional changes in the genes differentially expressed upon infection with WT HIV-1 (Figure 8a), compared to deletion of *vif* (Figure 8c), *vpu* (Figure 8d), and *nef* (Figure 8e). Vpr is a virion-encapsidated protein. To address whether virion encapsidated Vpr is sufficient to induce the transcriptional changes attributed to the presence of the *vpr* gene in the viral genome, we prepared samples of MDMs in parallel and infected those cultures with lentiviruses containing encapsidated Vpr, expressed from the packaging plasmid, pCMVΔR8.2 [34] (Vpr+) and lentiviruses using a packaging vector pCMVΔR8.2ΔVpr with *vpr* deleted (Vpr-). Comparison of Vpr+, containing encapsidated Vpr protein, to Vpr- lentiviral infection (green bars) to WT vs. mock (orange bars) reveals that virion encapsidated Vpr is sufficient to induce transcriptional changes in a majority of genes differentially expressed upon infection with WT HIV-1. Furthermore, these changes mirror an attenuated IFN response (Figure 8b) and most closely resemble a type-III response.

### 2.6. HIV-1 Vpu and Nef Induce Transcriptome Changes Related to Immune Functions

Vpu and Nef are widely reported to counter restriction factors that facilitate the ability of HIV-1 to evade the host immune system [35,36]. Protein interaction network mapping revealed that Vpu and Nef primarily affect clusters of genes that are involved in immune responses (Figure 9a,b, respectively). Transcriptional changes present in our RNA-seq dataset reveal enriched immune pathways involving chemokines and cytokines (Figure 9c,d and Appendix A). Specifically, interleukin-1 (IL-1), interleukin-7 (IL-7), and tumor necrosis factor alpha (*TNF-a*) signaling are enriched among both Vpu and Nef DEGs. Interestingly, *FOXM1*, a transcription factor with a prominent role in the cell cycle, is the most enriched transcription factor associated with Vpu-regulated genes suggesting a possible link between cytokine signaling and cell cycle control (Figure 9e and Appendix A). Nef-associated transcription factors include nuclear factor, erythroid 2 like 2 (*NFE2L2*), and androgen receptor (*AR*), a nuclear hormone receptor transcription factor. These observations suggest that antioxidant responses and hormone signaling are altered by Nef (Figure 9f and Appendix A). *RIPK2* and *IRAK2*, both significantly enriched among Vpr DEGs, are also significantly enriched among Vpu (Figure 9g and Appendix A) and Nef (Figure 9h and Appendix A) DEGs. This is not surprising given the broad role that these kinases play in immune signal transduction.

## 3. Discussion

The ability of HIV-1 infection to induce an IFN response has been a subject of controversy, with some reports indicating that HIV-1 induces a potent IFN response [2] and others suggesting that it does not [37]. The IFN response to HIV-1 infection in myeloid cells occurs in two stages: a rapid type-I response that occurs prior to reverse transcription that prevents subsequent infection in the cell [2,38,39]; and a more robust, second wave IFN response [38,39]. HIV-1 replication is inhibited by a type-I or -II IFN response due to the increased transcription of host antiviral factors (reviewed in [40]). However, the attenuated IFN response that we describe here may promote viral transcription through activation of NF-kB and IRF1 while avoiding the restrictions that would be imparted by a more robust IFN response. Because HIV-1 has evolved to manipulate and counteract the host IFN response through accessory proteins targeting components of the JAK/STAT IFN signaling pathway and induced viral restriction factors (reviewed in [36]), a modest IFN response does not necessarily preclude viral replication. Our dataset indicates that HIV-1 Vpr is responsible for induction of an attenuated IFN response and that encapsidated Vpr is sufficient to induce this response. Encapsidated Vpr is released very early in the HIV-1 live cycle and it is, therefore, possible that a Vpr-mediated early but modest IFN triggering is needed to augment viral fitness through enhanced transcription and/or immune evasion.

We also describe a correlation between transcriptomic signatures induced upon HIV-1 infection of macrophages and those induced in response to type-III IFN stimulation in our dataset and in a dataset from a previously published RNA-seq experiment [33]. This association was more apparent in a subset of cell cycle genes that are upregulated by Vif and/or Vpr, but downregulated after stimulation with type-I or -II IFNs. Induction of type-III IFN during HIV-1 infection has been noted but has received little attention [41]. Type-III IFNs (IFN-λ family) have been shown to be protective against HIV-1 infection [42,43,44,45], although type-III IFNs are less potent than type-I or -II IFNs [9]. IFN-λ imparts antiviral defense properties to mucosal boundaries that are less potent with slower but more sustained kinetics than type-I or -II IFNs (reviewed in [14]). Additionally, IFN-λ is not associated with the production of high levels of pro-inflammatory cytokines. Macrophages and dendritic cells are primary responders to IFN-λ gaining expression of the IFN-λ receptor, IFNLR1 upon differentiation from bone marrow derived monocytes to macrophages [46]. Our data suggests that HIV-1 infection induces a type-III IFN response in macrophages in addition to type-I and -II responses, consistent with observations in dendritic cells [41].

In addition to its effects on IFN signaling, we report here that HIV-1 infection of MDMs triggers transcriptional changes related to mitosis, DNA replication, and DNA damage responses. This observation is in agreement with a prior study [33] in which MDMs were infected with HIV-1_Bal_ and sorted into productively infected or bystander cells. The authors identified groups of genes related to mitosis and DNA replication in sorted, productively infected MDMs 36 h after infection that were similar to those highlighted in the current study. In addition, we report that Vpr and Vif are required for upregulation of cell cycle regulatory genes. We further characterized this group of genes being potently downregulated upon stimulation with type-I and type-II IFNs, which are thought to be produced as a consequence of HIV-1 infection. The functional significance of increased transcription of mitotic genes as a result of HIV-1 infection in non-cycling cells remains unclear. Others have reported that a G1-like state in macrophages provides an environment more conducive to reverse transcription due to higher concentrations of dNTPs subsequent to inactivating phosphorylation of sterile alpha motif and HD domain containing protein 1 (*SAMHD1*) [47,48]. We do uncover changes in gene transcription related to G1/S specific pathways as a result of Vif and Vpr. However, mitotic genes are not expected to play a significant role in G1 processes. The requirement of the microtubule network for HIV capsid trafficking and for cell-to-cell infection of HIV-1 is well documented (reviewed in [32,49]). HIV-1 manipulation of the microtubule network is highlighted by a study in MT4 cells describing Vpr localization to the centrosome, targeting and degrading CP100, and a resulting increase in microtubule nucleation [50]. The consequences of the Vpr induced increase in microtubule nucleation on HIV-1 infection are unknown.

Both Vpr and Vif induce G2/M cell cycle arrest in cycling cells such as CD4 T cells [51,52,53]. One leading explanation regarding the selection of G2/M cell cycle arrest by HIV-1 and other viruses is the notion that a prolonged G2 phase enhances viral fitness by setting up a state that enhances proviral transcription due to intrinsic properties of the viral long terminal repeat, which is a G2/M responsive promoter [54,55]. Vif induces G2/M cell cycle arrest by targeting protein phosphatase 2A (*PP2A*) subunit B56 (*PP2R5*). *PP2A/PPP2R5* complexes are known to regulate multiple G2/M checkpoints involving cyclin dependent kinase 1 (*CDK1*) activation, altered regulation of Aurora kinases, and premature centromere dissociation (reviewed in [53]). Processes related to centromere assembly, Aurora kinase signaling, and CDK1 activation are highly enriched among the subset of genes in that are upregulated by Vif and downregulated by type-I and -II IFNs. In addition, Vif is necessary for the degradation of apolipoprotein B mRNA editing enzyme catalytic polypeptide (*APOBEC*) cytidine deaminases, which are potent anti-retroviral proteins. It does so by hijacking core-binding factor subunit b (*CBFβ*) and cullin-5 (*CUL5*) to form a stable *CBFβ-CUL5-ELOBC-Vif* E3 ligase complex resulting in subsequent ubiquitination and degradation of APOBECs by the proteasome [56,57]. It is plausible that Vif-mediated degradation of PPP2R5 drives our observed cell cycle gene transcriptomic signature. The relationship of these genes with the response to type-I and -II IFN stimulation is less clear. It is possible that *CDK1* downregulation upon IFN stimulation is necessary for negative regulation of IFN signaling. Interestingly, *CBFβ* was recently identified as an upstream regulator of IFN-β and IFN-λ [41].

The mechanism through which Vpr mediates G2/M cell cycle arrest is less clear. This may be due to the promiscuous nature of Vpr-mediated proteome remodeling highlighted in a study in which the researchers identified 38 proteins with Vpr-mediated degradation requiring the *DCAF1/DDB1/CUL4* E-3 ligase complex [58], the ubiquitin ligase complex required for Vpr induced G2/M cell cycle arrest (reviewed in [52]). Independent of this study, reports implicate *MUS81*, *MCM10,* and *CCDC137* as playing a role in G2/M cell cycle arrest [59,60,61]. Kinases, such as *CDK1* and *AURKB,* and transcription factor *E2F4* related to G2/M checkpoints are highly enriched among our dataset of Vpr-regulated genes. However, the mechanism driving these changes will require further clarification.

Our work supports the notion that HIV-1 induces a modest interferon response in macrophages upon infection and identifies accessory protein Vpr as the predominant driver of this response. This work also complements previous studies [9,33] that document upregulation of cell cycle-related genes upon infection of macrophages with HIV-1 and identifies accessory protein Vif as the predominant driver of this response.

## 4. Materials and Methods

Isolation of healthy donor PBMC. Healthy donors 18 years old and older were recruited for this study under the University of Utah Institutional Review Board (IRB) protocol 67637. Written informed consent was obtained from all donors. Whole blood was obtained by peripheral phlebotomy, and peripheral blood mononuclear cells (PBMC) were isolated using a Lymphoprep density gradient (Stemcell Technologies, Vancouver, Canada).

Generation and infection of MDM. CD14+ monocytes were isolated by positive selection with magnetic beads (Miltenyi Biotec, San Diego, CA, USA). Monocytes underwent differentiation to macrophages as previously described [22]. MDM were infected with 250 ng of HIV-1-BAL-HSA or with a clone with a single accessory gene deletion as determined by p24 enzyme-linked immunosorbent assay (ELISA) (Zeptometrix, Buffalo, NY, USA) 6 h. Cells were washed twice with fresh medium to remove unbound virus. Infection was quantified via flow cytometry at 48 h post-infection.

Generation of mutant HIV-1_Bal_ clones. HIV-1-BAL-HSA (HIV-1_Bal_) was used to create the following mutants: ΔVpr, ΔVif, ΔVpu, ΔNef. All mutations were confirmed by sequencing. We constructed the ΔVpr clone by destroying the unique EcoR1 restriction site within the *vpr* locus. A previous report verifies that this mutation ablates transcription of Vpr (Figure 5 in [62]). ΔVif was constructed by cloning the Sph1-Sal1 fragment from HIV-1_Bal_ into Puc19, generating an Xba site within the vif locus at codons 37–38 using PCR to generate the mutant (QuikChange Lightning, Agilent, Santa Clara, CA, USA)). The generated Xba1 site was then cleaved and blunted using Klenow large fragment (New England Biolabs, Inc., Ipswich, MA, USA) and cloned into HIV-1_Bal_ using Sph1 and Sal1 to replace the WT *vif* ORF. The ΔVpu mutant was generated in a manner similar to the ΔVif mutant with the following exceptions: the Sal1-BamH1 HIV-1_Bal_ fragment containing *vpu* was cloned into Puc19. An Xba1 site was generated in *vpu* ORF amino acid positions 3–5. The Xba site was then cleaved, blunted, and then cloned into HIV-1_Bal_ using Sal1 and BamH1 to replace the WT *vpu* ORF. The ΔNef mutant was generated in a manner similar to the ΔVif mutant with the following exceptions: the Sph1-BamH1 HIV-1_Bal_ fragment containing *nef* was cloned into Puc19. An Xho1 site present in *nef* ORF amino acids 313–314 was cleaved and blunted. The Xba site was then cleaved, blunted, and cloned into HIV-1_Bal_ using Sal1 and BamH1 to replace the WT *nef* ORF.

Generation of viruses. Replication-competent viruses (HIV-1_Bal_), including those containing mutations in accessory genes, were generated using calcium phosphate-mediated transfection of HEK293T cells with a single plasmid (pNL-43-BAL-IRES-HSA) courtesy of Michel Tremblay (Centre Hospitalier de l’Université Laval) or accessory gene mutants described above. The transfection medium was removed 6 h later, cells were cultured over 1 day, and virus-containing supernatants were removed 24 h post-transfection. All viruses were quantified using p24 ELISA (Zeptometrix) and stored at −80 °C until further use.

RNA-seq sample preparation. MDM were generated as previously described and stimulated with 25 ng/mL of the indicated IFNs. Total RNA was isolated 18 (IFN stimulation study) and 24 h (infection with accessory gene deletion study) following stimulation using RNeasy minikit (Qiagen, Hilden, Germany). Intact poly(A) RNA was purified from total RNA samples (100–500 ng) with oligo(dT) magnetic beads, and stranded mRNA sequencing libraries were prepared as described using the Illumina TruSeq Stranded mRNA Library Preparation kit (catalog no. RS-122-2101 and RS-122-2102). Purified libraries were qualified on an Agilent Technologies 2200 TapeStation using a D1000 ScreenTape assay (catalog no. 5067-5582 and 5067-5583). The molarity of adapter-modified molecules was defined by quantitative PCR using the Kapa Biosystems Kapa Library Quant kit (catalog no. KK4824). Individual libraries were normalized to 10 nM, and equal volumes were pooled in preparation for Illumina sequence analysis. Sequencing libraries (25 pM) were chemically denatured and applied to an Illumina HiSeq v4 single-read flow cell using an Illumina cBot system. Hybridized molecules were clonally amplified and annealed to sequencing primers with reagents from an Illumina HiSeq SR cluster kit v4-cBot (catalog no. GD-401-4001). Following transfer of the flow cell to an Illumina HiSeq 2500 instrument (HCSv2.2.38 and RTA v1.18.61), a 50-cycle single-read sequence run was performed using HiSeq SBS kit v4 sequencing reagents (catalog no. FC-401-4002).

RNA data analysis. The human GRCh38 genome and gene feature files were downloaded from Ensembl release 94 and the reference database was created using STAR version 2.6.1b with splice junctions optimized for 50 base pair reads [63]. Reads were trimmed of adapters using cutadapt 1.16 (Marcel 2011) and aligned to the reference database using STAR in two pass mode to output a BAM file sorted by coordinates. Mapped reads were assigned to annotated genes using feature Counts version 1.6.3 [64] and differentially expressed genes were identified using a 5% false discovery rate with DESeq2 version 1.20.0 [65]. Unmapped reads that matched at least 31 bases in four virus genes (allowing 1 base mismatch) were counted using Seal from BBtools.

Gene ontogeny. Heat maps we generated using the NG-CHM Builder [66]. Venn diagrams were generated using the following tool: https://bioinformatics.psb.ugent.be/webtools/Venn/ accessed on 2 November 2021. Enriched pathways, transcription factors, and kinases were identified using Enrichr [67,68,69]. The following databases within Enrichr were selected for analysis: ENCODE and ChEA Consensus transcription factors, Bioplanet 2019 human pathways, ARCHS4 for genes co-expressed with kinases.

Principal component analysis was performed using GraphPad Prism Software version 9.2.0. STRING analysis was performed using Cytoscape version 3.8.2 to generate protein interaction networks

RT-qPCR. RNA was isolated from MDMs using a commercial kit (Qiagen RNeasy) and quantified using nanodrop. Samples were prepared using commercial one step RT-qPCR kit (Qiagen QuantiTect SYBR^®^ Green RT-PCR Kit) and were run on a Roche LC480 light cycler. The following primers were used for quantification: *IDO1* primer 1: TCATGGAGATGTCCGTAAGG, *IDO1* primer 2: GCCAAGACACAGTCTGCATA, *PLK1* primer 1: AATTACATAGCTCCCGAGGTG, *PLK1* primer 2: AGCCAGAAGTAAAGAACTCGTC. *GAPDH1* primer 1: AGCCTCAAGATCATCAGCAATGCC, *GAPDH1* primer 2: TGTGGTCATGAGTCCTTCCACGAT. *KIF20A* primer 1: GAACCTGCTATCAGACTGCT *KIF20A* primer 2: TGATCTTCCTGTCGTTCCAAC. CT values were converted to gene expression using the 2^(−ΔΔCT) method. Expression of *IDO1*, *PLK1,* and *KIF20A* were normalized to expression of housekeeping gene *GAPDH1*.

Western blotting. MDMs were lysed in NTEN buffer supplemented with phosphatase and protease inhibitors (Sigma, St. Louis, MO, USA). Protein lysates were clarified by centrifugation and quantified using BCA assay (ThermoFisher, Waltham, MA, USA). A total of 25 μg of lysates were loaded onto SDS-PAGE gels (Criterion, Bio-Rad, Hercules, CA, USA), separated, then transferred onto 0.45 μM pore size PVDF membranes using a semi-dry transfer apparatus (Bio-Rad). Membranes were blocked using 4% BSA and cut into sections for incubation with primary antibodies. Membranes were incubated with the following antibodies: Vif, Nef, and Vpu from the NIH AIDS Reagent Repository, or Hsp90 (Cell Signaling Technology, Danvers, MA, USA) for 18 h at 4 °C. The membranes were washed, then incubated with HRP conjugated secondary antibodies, developed using ECL reagents (ThermoFisher), and exposed using a Biorad Western Blot Imager.

Flow cytometry. MDMs were detached using accutase (Gibco, Carlsbad, CA, USA) for 2 h at 37 °C. Cells were washed with PBS and then stained for 30 min at 4 °C using a violet viability dye (Life Technologies, Carlsbad, CA, USA) and a AF647 conjugated mouse CD24 antibody (R&D Systems, Minneapolis, MN, USA) in the presence of an FC blocking antibody (Miltenyi Biotec). Cells were then washed with PBS and fixed in the presence of 0.5% paraformaldehyde for 30 min at 4 °C. Viability and surface expression of CD24 were quantified using Pacific blue and AF647 fluorescence detected on a BD LSRFortessa^TM^ X-20 flow cytometer. Fluorescence was analyzed using FlowJo Software.

## Figures and Tables

**Figure 1 pathogens-11-00163-f001:**
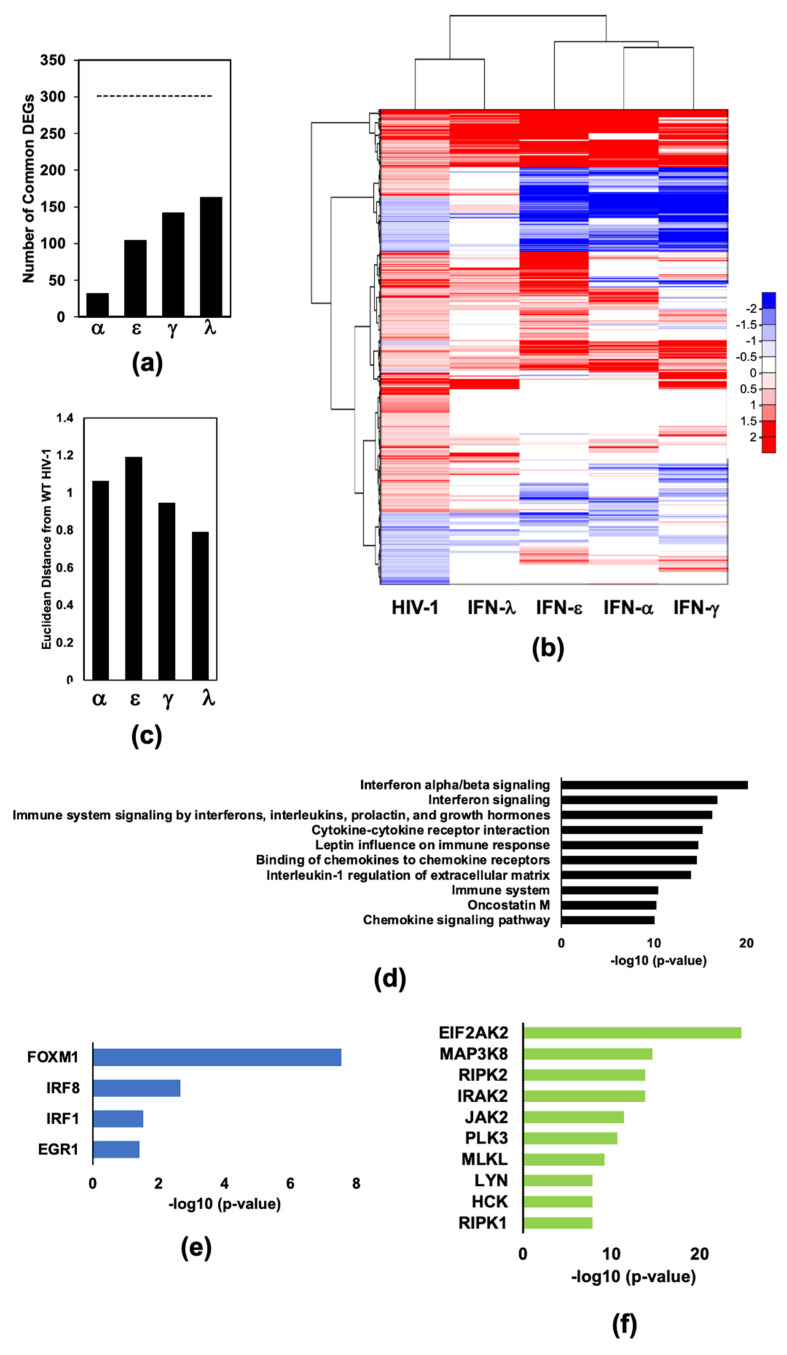
HIV-1 infection induces a type-III IFN-like response in macrophages. MDMs were either mock infected, infected with HIV-1_Bal_ (HIV-1) with RNA isolated 24 h post infection, or left untreated or treated with 25 ng/mL IFN-α, IFN-ε, IFN-γ, or IFN-λ with RNA isolated 18 h post-infection [9,24]. (**a**) Bar graph showing the percentage of differentially expressed genes (DEGs) upon infection of MDMs with HIV-1_Bal_ that are also differentially expressed upon stimulation with IFN-α, IFN-ε, IFN-γ, or IFN-λ. (**b**) Heat map clustering of RNA-seq DEGs from the following comparisons (from left to right, HIV-1_Bal_ vs. mock infection, IFN-λ vs. untreated, IFN-ε vs. untreated, IFN-α vs. untreated, IFN-γ vs. untreated (P_adj_ < 0.05, log2FC < 0.6 or > 0.6)). (**c**) Euclidean distance of each IFN transcriptome change from HIV-1_Bal_ induced transcription changes. Distances were normalized to the average of four computed distances. (**d**) Top 10 Bioplanet pathways of genes enriched among HIV-1 induced DEGs with *p*-value < 0.05. (**e**) Four ENCODE and ChEA transcription factors from ChIP-X of genes enriched among HIV-1 induced DEGs with *p*-value < 0.05. (**f**) Top 10 ARCHS4 kinases of genes enriched among HIV-1 induced DEGs with *p*-value < 0.05.

**Figure 2 pathogens-11-00163-f002:**
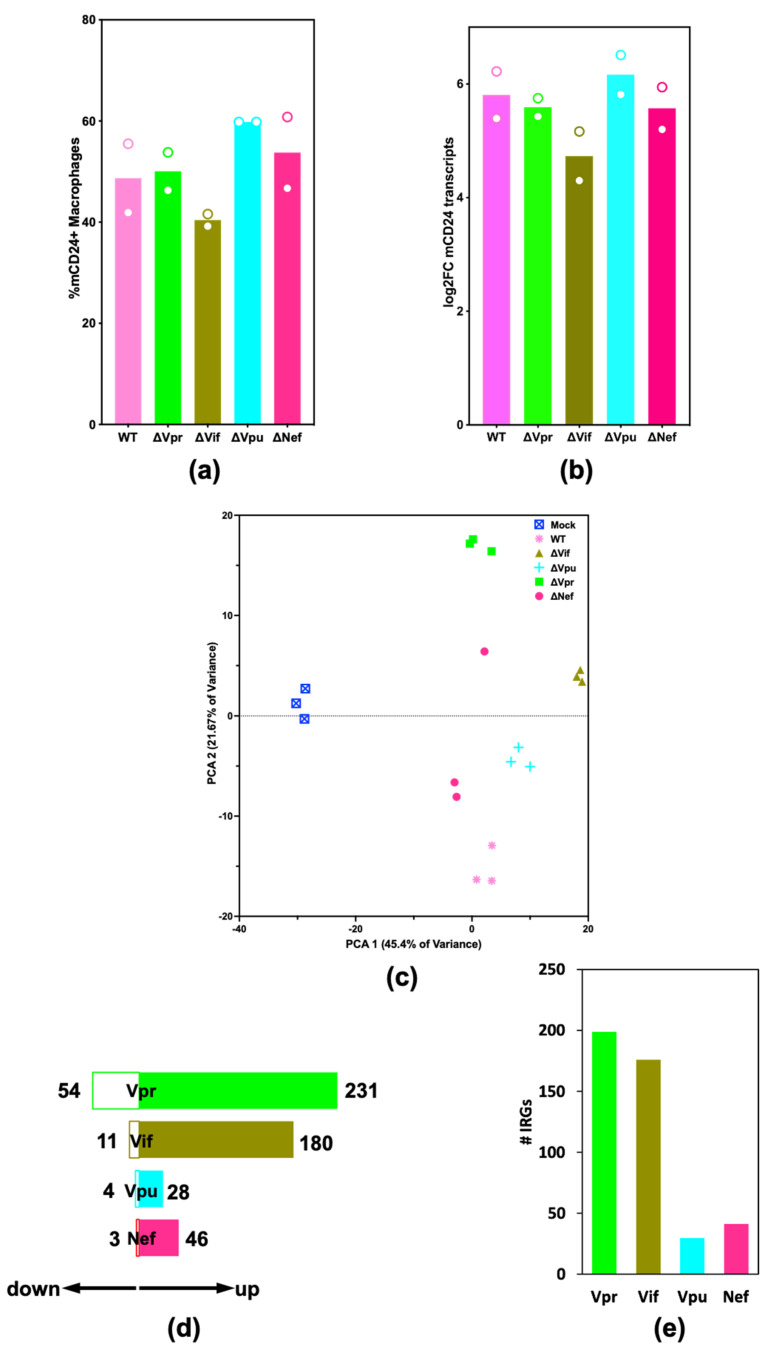
HIV-1 Vif and Vpr significantly remodel the transcriptomes of macrophages. MDMs were infected with HIV-1_Bal_ or mutants HIV-1_Bal_ΔVpr (ΔVpr), HIV-1_Bal_ΔVif (ΔVif), HIV-1_Bal_ΔVpu (ΔVpu), HIV-1_Bal_ΔNef (ΔNef). RNA was isolated 24 h after infection and a sample of cells was taken from each for flow cytometry quantification of cell surface murine CD24 (mCD24). (**a**) Virally encoded mCD24 transcripts were measured. Bars represent the average log2FC expression from these mCD24 transcripts and each is depicted as separate points in the plot. (**b**) Percentage of MDMs productively infected with HIV-1_Bal_ determined using flow cytometry detecting virally encoded mCD24 (**c**) Principal component analysis (PCA) plot comparing the top 500 most variable transcripts from three replicates of MDMs either mock infected or infected with HIV-1_Bal_ viruses: wild-type, ΔVpr, ΔVif, ΔVpu, ΔNef. (**d**) Number of upregulated (filled bars) and downregulated (empty bars) DEGs (P_adj_ < 0.05, log2FC < 0.6 or > 0.6) resulting from comparisons between WT and mutant viral infections. (**e**) Number of IRGs differentially expressed upon deletion of *vpr*, *vif*, *vpu,* and *nef* from the HIV-1 genome.

**Figure 3 pathogens-11-00163-f003:**
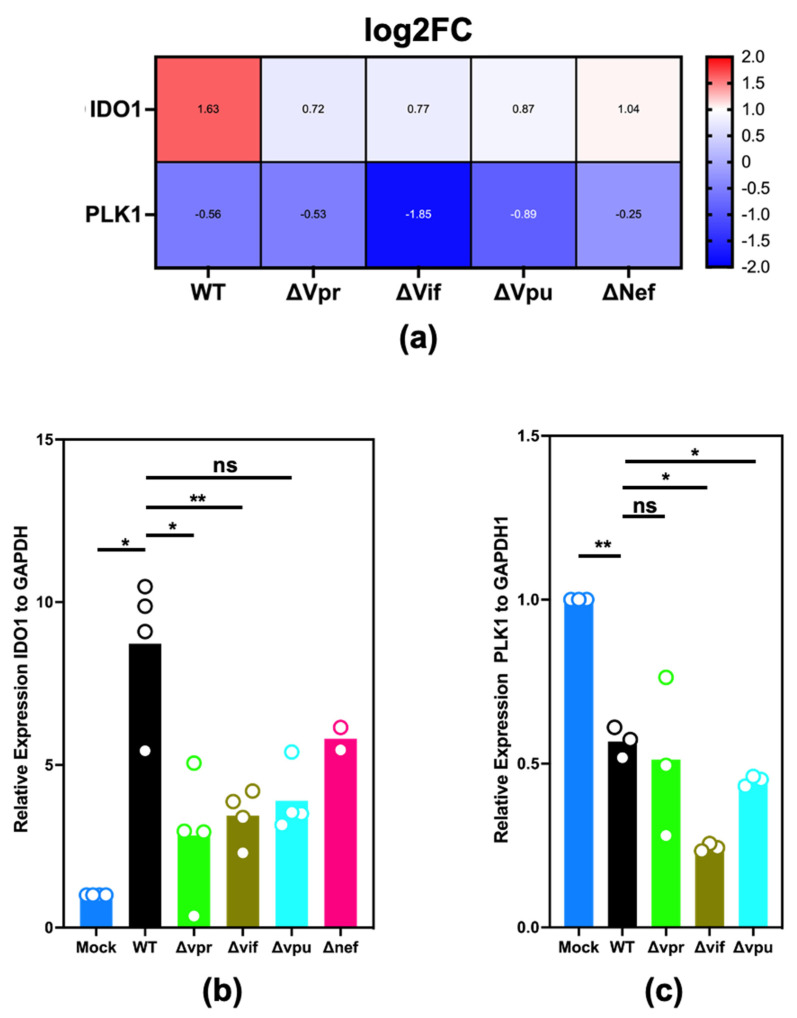
RT-qPCR validation of *IDO1* and *PLK1* differential expression. RNA extracts from MDMs subjected to RNA-seq were subjected to RT PCR. (**a**) Heatmap depicting *IDO1* and *PLK1* expression levels (log2FC from mock infection) from RNA-seq. (**b**) *IDO1* expression from three replicate RNA samples that were also subjected to RNA-seq. (**c**) *PLK1* expression from three replicate RNA samples that were also subjected to RNA-seq. Gene expression was determined using the ΔΔCt method. KIF20A expression levels are normalized to untreated samples and then to GAPDH expression. *p*-values computed using a two-tailed paired Student’s t test and significance assigned as follows: *: *p*-value < 0.05; **: *p*-value < 0.01; ns: *p*-value > 0.05.

**Figure 4 pathogens-11-00163-f004:**
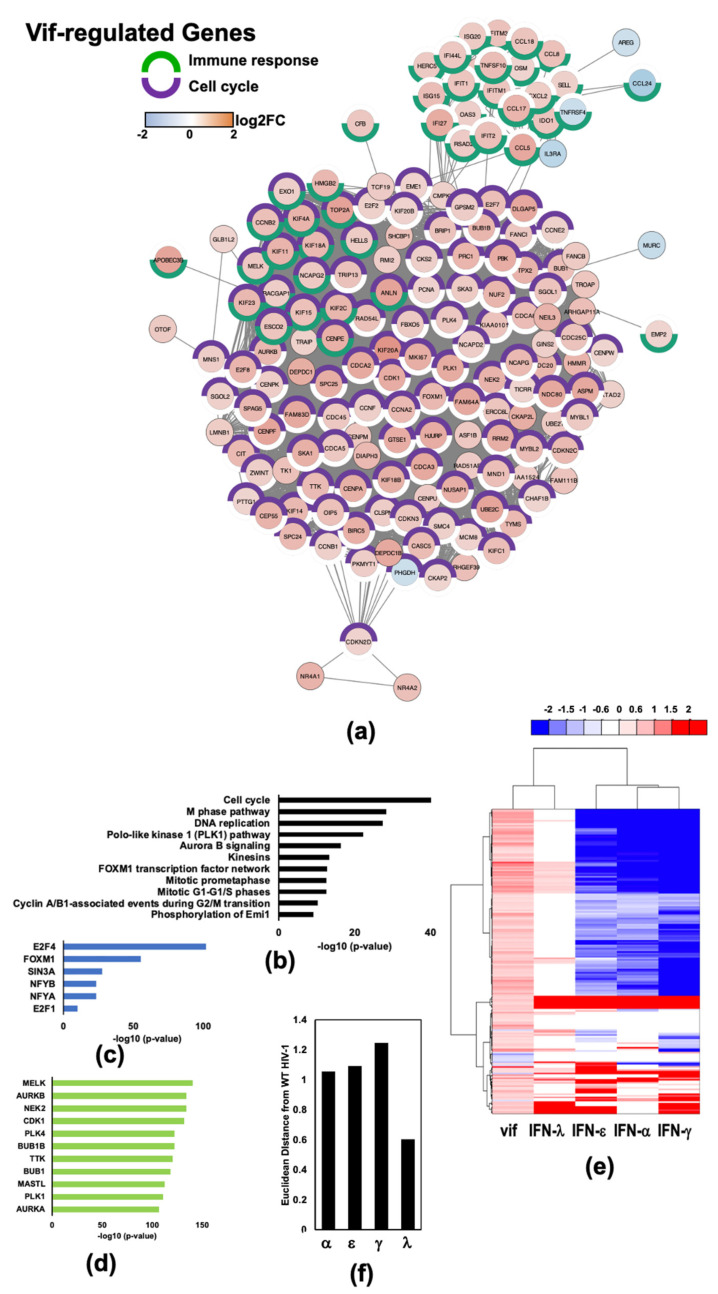
HIV-1 Vif induces a large number host transcriptome changes related to mitosis. (**a**) Protein interaction network (PIN) of Vif induced DEGs generated using Cytoscape^TM^ software to perform STRING analysis. DEGs that are < or > 0.6 logFC and padj < 0.05 when comparing transcriptomes from WT to ΔVif infected MDMs were included. Unclustered genes were removed. (**b**) Top 11 enriched Bioplanet pathways from 191 DEGs using criteria described in (**a**). (**c**) Top 6 enriched ENCODE and ChEA transcription factors from ChIP-X from 191 DEGs using constraints described in (**a**). (**d**) Top 11 enriched ARCHS4 kinases from 191 DEGs using constraints described in (**a**). (**e**) Heat map comparison of Vif-regulated genes to DEGs from 25 ng/mL IFN-α, IFN-ε, IFN-γ, IFN-λ-stimulated MDMs [9,24]. (**f**) Euclidean distance of each IFN transcriptome change from HIV-1_Bal_ induced transcription changes. Distances were normalized to the average of four computed distances.

**Figure 5 pathogens-11-00163-f005:**
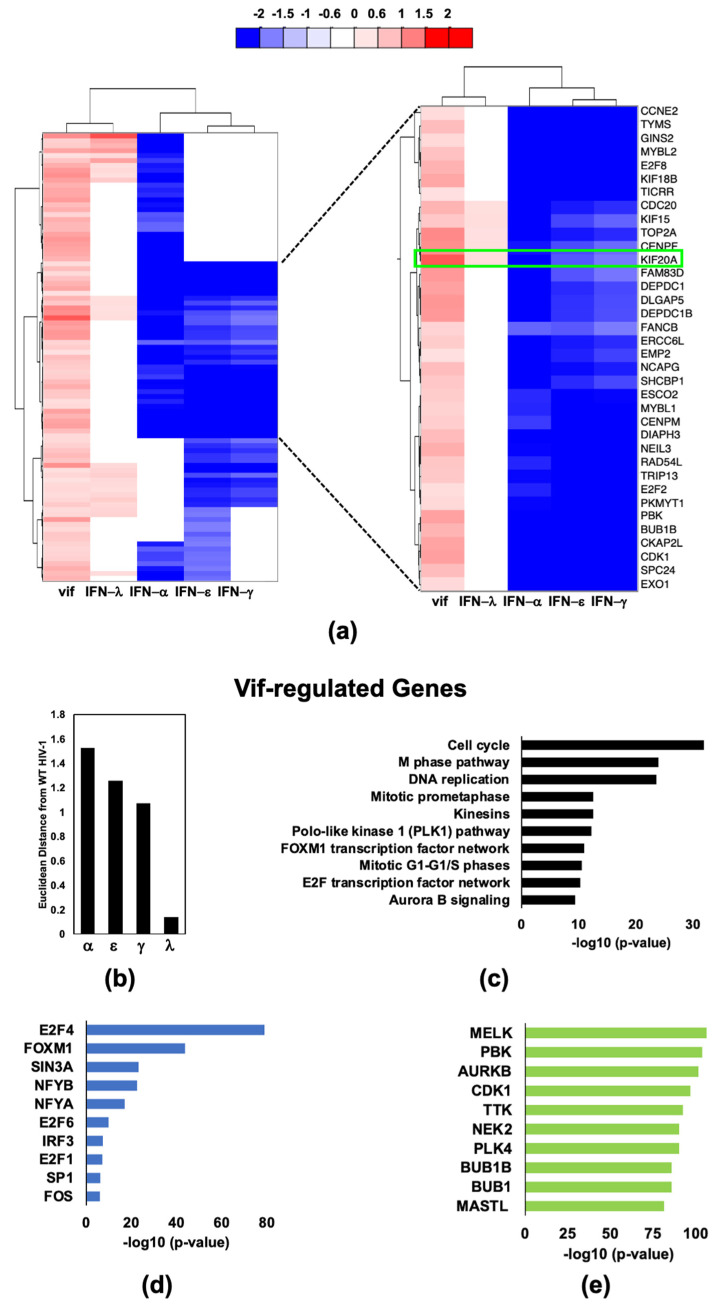
HIV-1 Vif upregulates a cluster of genes that are potently downregulated by Type-1 and -II IFNs (**a**) Heat map comparison of WT vs. mock with WT vs. ΔVif DEGs (< or >0.6 logFC and padj < 0.05). (**b**) Euclidean distance of each IFN transcriptome change from HIV-1_Bal_ induced transcription changes. Distances were normalized to the average of four computed distances. (**c**) Top 10 enriched Bioplanet pathways of genes included in the panel a heatmap. (**d**) Top 10 enriched ENCODE and ChEA transcription factors from ChIP-X of genes included in the panel a heatmap (**e**) Top 10 enriched ARCHS4 kinases of genes included in the panel a heatmap.

**Figure 6 pathogens-11-00163-f006:**
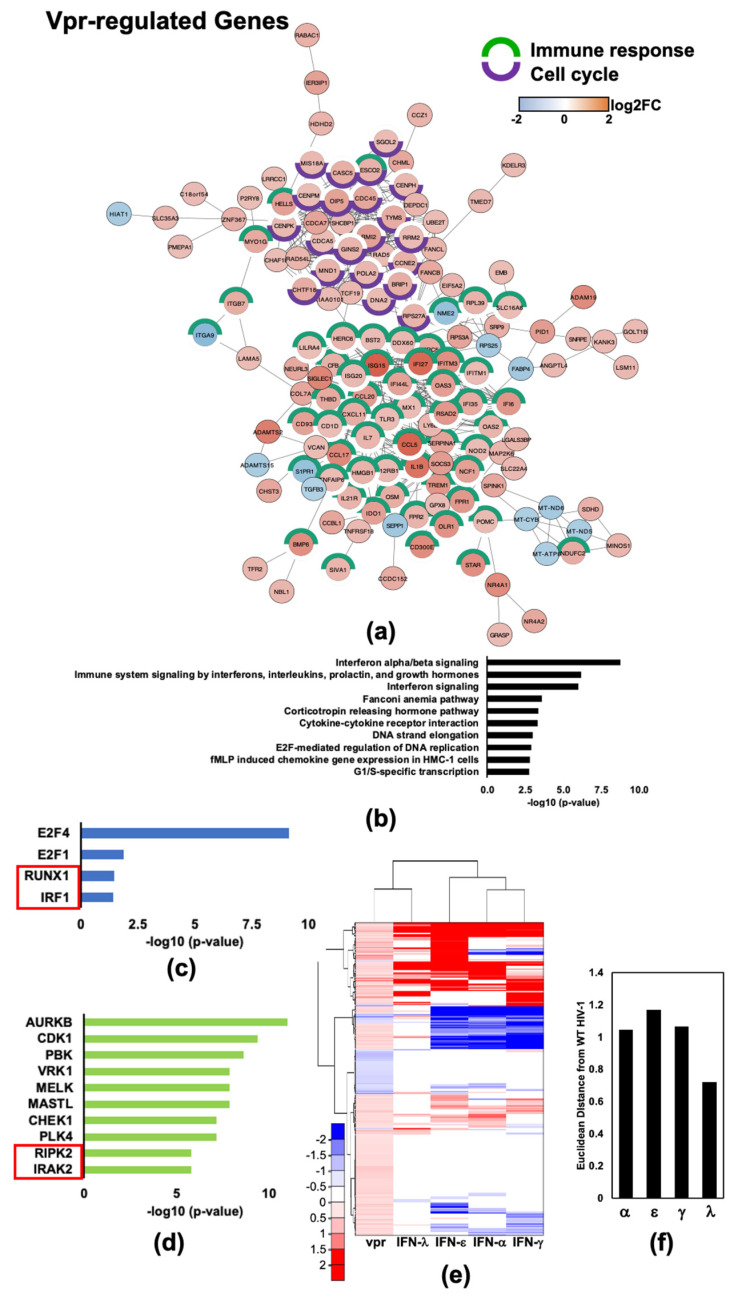
HIV-1 Vpr remodels the macrophage transcriptome to induce changes in IFN-regulated genes and genes involved in DNA replication. (**a**) Protein interaction network (PIN) of genes that were differentially expressed by > or < 0.6 log2FC and with *p*-value < 0.05 when comparing transcriptomes of WT vs. Δvpr infected MDMs. (**b**) Top 10 enriched Bioplanet pathways. (**c**) Top 4 enriched ENCODE and ChEA transcription factors from ChIP-X. Red boxes indicate transcription factors that are not found in the list of significantly enriched kinases in the Vif-regulated gene set. (**d**) Top 10 enriched ARCHS4 kinases. Red boxes indicate transcription factors that are not found in the list of significantly enriched kinases in the Vif-regulated gene set. (**e**) Heat map comparison of Vpr-regulated genes to genes differentially regulated upon stimulation with IFN-α, IFN-ε, IFN-γ, and IFN-λ (dataset #1) [9,24]. (**f**) Euclidean distance of each IFN transcriptome change from HIV-1_Bal_ induced transcription changes. Distances were normalized to the average of four computed distances.

**Figure 7 pathogens-11-00163-f007:**
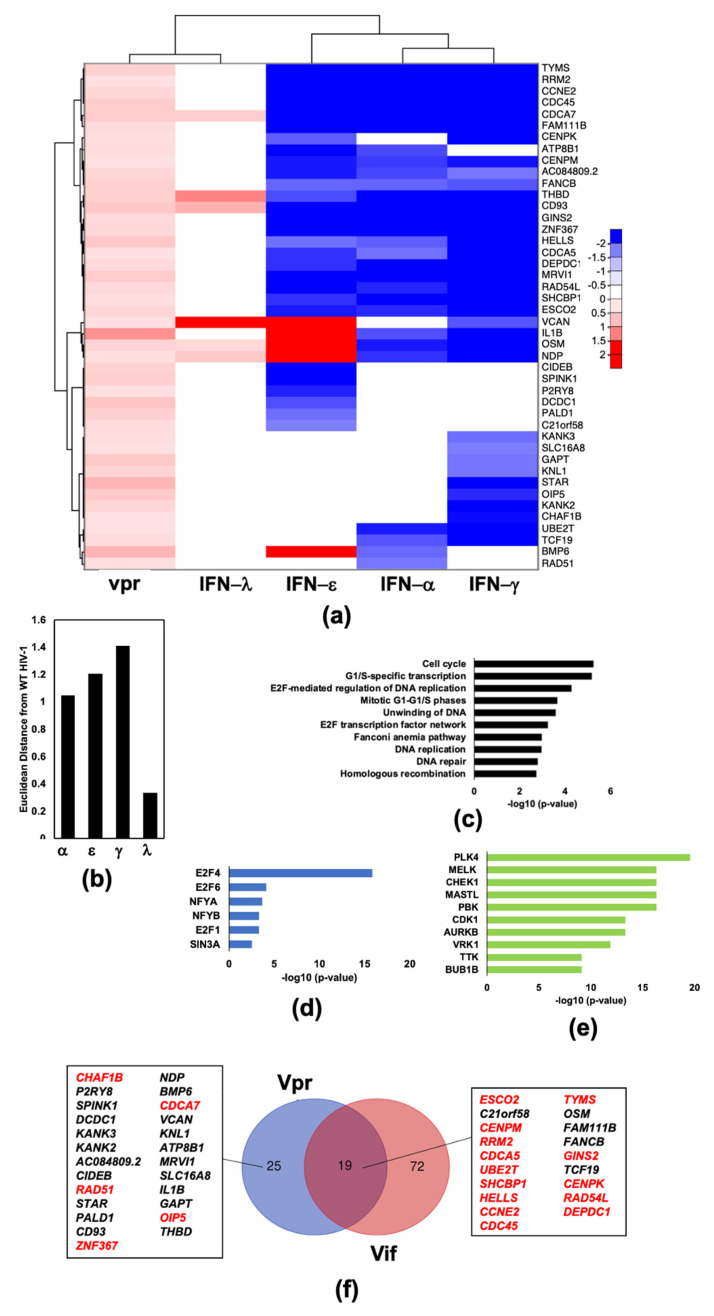
HIV-1 Vpr upregulates a cluster of genes that are potently downregulated by type-I and -II IFNs. (**a**) Heat map comparison of Vpr upregulated IRGs with genes differentially regulated after IFN stimulation. (**b**) Euclidean distance of each IFN transcriptome change from HIV-1_Bal_ induced transcription changes. Distances were normalized to the average of four computed distances. (**c**) Top 10 enriched Bioplanet pathways of genes included in the panel a heatmap. (**d**) The 6 significantly (*p*-value < 0.05) enriched ENCODE and ChEA transcription factors from ChIP-X of genes included in the panel a heatmap. (**e**) Top 10 enriched ARCHS4 kinases of genes included in the panel a heatmap. (**f**) Venn diagram comparing Vpr and Vif upregulated genes that are potently downregulated by type-I and -II IFNs. Genes known to be regulated by transcription factor E2F4 are labeled red.

**Figure 8 pathogens-11-00163-f008:**
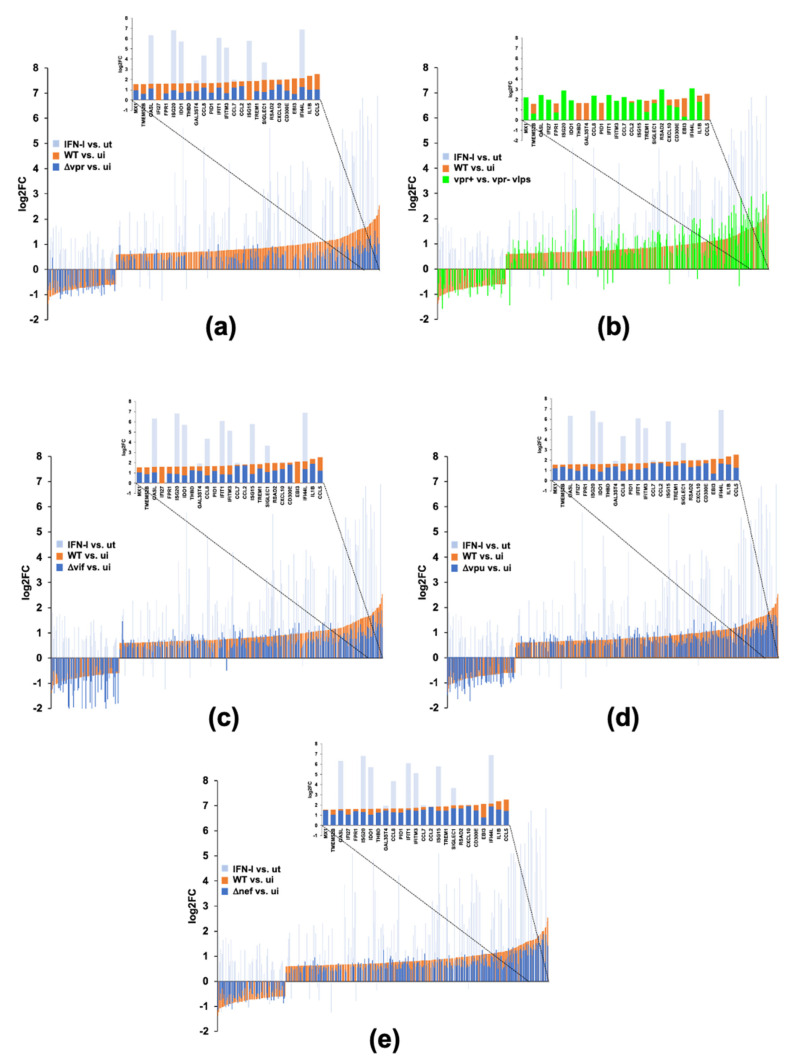
HIV-1 Vpr contributes the most significantly to changes in the transcriptome from IFN-related genes relative to Vif, Vpu, and Nef. Overlay plots comparing DEGs (P_adj_ < 0.05, log2FC < 0.6 or > 0.6) resulting from IFN-λ vs. untreated [9,24] (light grey bars) and WT HIV-1_Bal_ infection of MDMs (orange bars) to the following: (**a**) ΔVpr vs. mock (blue bars), (**b**) Vpr- vs. Vpr+ lentiviral infection (green bars), (**c**) ΔVif vs. mock (blue bars), (**d**) ΔVpu vs. mock (blue bars), (**e**) ΔNef vs. mock (blue bars).

**Figure 9 pathogens-11-00163-f009:**
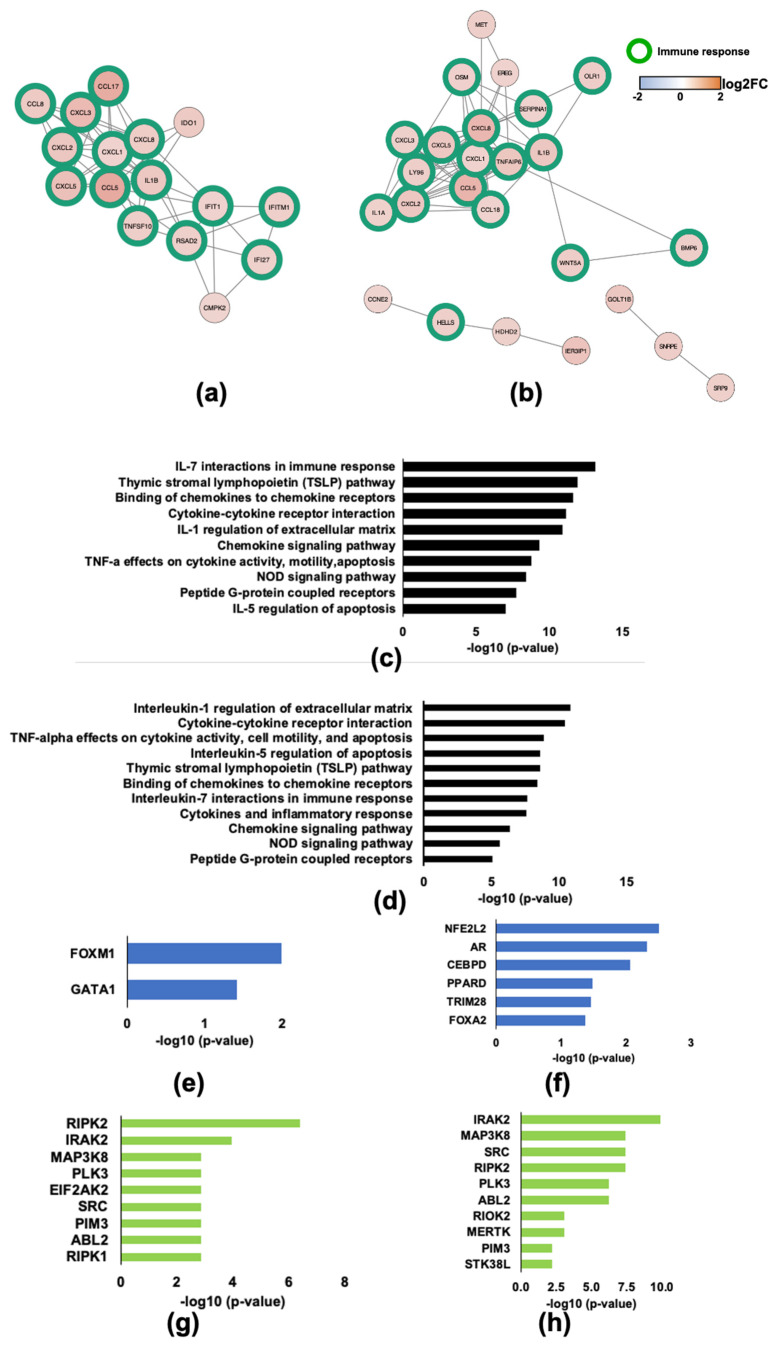
HIV-1 Vpu and Nef transcriptome changes are enriched among genes related to cytokine and chemokine signaling and processes. (**a**) Protein interaction network (PIN) of Vpu-regulated DEGs. (**b**) Protein interaction network (PIN) of Nef-regulated DEGs. (**c**) Top 10 enriched Bioplanet pathways in Vpu-regulated genes. (**d**) Top 10 enriched Bioplanet pathways in Nef-regulated genes. (**e**) Two significantly enriched transcription factors in Vpu-regulated genes. (**f**) Six significantly enriched transcription factors in Nef-regulated genes. (**g**) Top 8 enriched ARCHS4 kinases in Vpu-regulated genes. (**h**) Top 9 enriched ARCHS4 kinases in Nef-regulated genes.

## Data Availability

The data supporting this study are available in the Appendix A at https://www.mdpi.com/article/10.3390/pathogens11020163/s1.

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
