# Peer review of "HIV-1 Accessory Proteins Impart a Modest Interferon Response and Upregulate Cell Cycle-Related Genes in Macrophages"

_pathogens, 2022, doi:10.3390/pathogens11020163_

Round 1

Reviewer 1 Report

While it is known type I interferon responses are critical for establishing an anti-viral state to combat HIV-1 infection, less is known about the impact of other interferons such as type II and III on HIV-1 infection

The authors also tested whether innate immune responses from macrophages directed towards HIV-1 resembles interferon regulated gene (IRG) signatures, and presumably whether these are reminiscent of type I, II, or III responses

Finally, the authors tested if HIV-1 accessory genes can interfere with type I and II IFN generation.

Overall, this manuscript requires the following revisions:

Figure 1 :

Their description of the experiments was not very clear.  

In the figure legend, the authors state that RNA was isolated 18 hours post-infection. It’s not written explicitly, but are they stating these cells are infected with HIV-1BAL first, then treated/untreated with interferon? If not, this is a typo that needs correcting.

The authors extracted RNA from cells in the infection condition 24 hours post-infection. Since they used replication competent virus, is this enough time for HIV-1 to induce the interferon responses and then for those interferon responses to induce subsequent signalling in a paracrine/autocrine manner to eventually stimulate/repress interferon sensitive genes?

The authors do not mention what MOI they used during the infection. Since they are using replication competent virus, how are they ensuring infection at the correct MOI such that superinfection does not occur?

My major issue with this figure is that they authors widely discuss about how the Type I and II interferon responses are much more potent than Type III interferon responses but do not set up a condition where MDMs are cultured with different combinations of Type I/II and Type III interferons to see what the overall net effect is? Is it possible the Type III interferon effect is dampened or overpowered by the presence of Type I/II interferons?

The authors should clarify that the DEGs presented in panels C, D and E are strictly from the HIV-1 infected MDMs and are not the DEGs in common with the interferon treatments. This is a problem because they mislabeled which panel was the heat map in the figure legend, which actually would change the interpretation of the results

The authors should quantify that the HIV-1 infection condition is “most like Type III”, as they state in the text.

Figure 2:

Some  issues with the presented WB. 

The Western blots need to include MW markers

The Nef blot has a massive water bubble and should be repeated

The authors state that they couldn’t blot for Vpr as no antibody exists for it. Blots for Vpr have been reported in the litterature

The Figure legend states that the Nef blot is panel B when it’s actually C

In Figure 2a and b, the authors should provide error bars, an indication of whether variance is represented as SEM or SD, and include statistical testing.

With respect to their flow cytometry experiments, the authors should present their gating strategies . Was the respective gating performed with an FMO control? Gating strategies were not mentioned in the methods.

Figure 3 

The authors should provide a rationalization for looking at IDO1 and PLK1.

Here the authors use MDMs also infected with Δall accessory genes (DDDD) virus. It is unclear why this virus was not used in prior experiments as a functional negative control?

Authors need a statistical analysis for these experiments and while they acknowledge their RNA-seq and RT-qPCR experiments for IDO1 don’t agree with each other they need to explain this.

Figure 4 and 5

The authors acknowledge that 50% of the Vif-induced upregulated genes are actually potently downregulated by Type I and Type II interferons. Similar results were observed for Vpr. They found most of these genes were cell cycle regulators and resembles a Type III interferon response. But considering Type I and II interferons surely exist in vivo, then this may have very little significance. This should be discussed.

Figure 6

It is unclear how the protein interaction network maps were generated. Are these DEGs? It says they are in the figure legend, but doesn’t provide that information in the text

Author Response

Figure 1: “Their description of the experiments was not very clear”. Response. We have inserted additional clarifications regarding experimental design.

“In the figure legend, the authors state that RNA was isolated 18 hours post-infection. It’s not written explicitly, but are they stating these cells are infected with HIV-1BAL first, then treated/untreated with interferon? “. Response. Sorry about this confusion. For interferon treatments, RNA was isolated at 18 hours post treatment. For viral infections, RNA was isolated at 24 hours post-infection. This has been clarified in the text.

“The authors extracted RNA from cells in the infection condition 24 hours post-infection. Since they used replication competent virus, is this enough time for HIV-1 to induce the interferon responses and then for those interferon responses to induce subsequent signaling in a paracrine/autocrine manner to eventually stimulate/repress interferon sensitive genes?” Response. This is a great point. We aimed to characterize the transcriptome response to HIV-1 infection at a time late enough to detect post integration HIV-1 induced changes but early enough to capture the early post integration changes and avoid complications due to subsequent signaling.

“The authors do not mention what MOI they used during the infection”.Response: As stated in the M&M section, 250ng of each virus was used to infect 600,000 cells.

“My major issue with this figure is that they authors widely discuss about how the Type I and II interferon responses are much more potent than Type III interferon responses but do not set up a condition where MDMs are cultured with different combinations of Type I/II and Type III interferons to see what the overall net effect is? Is it possible the Type III interferon effect is dampened or overpowered by the presence of Type I/II interferons?” Response: Culturing the cells in combinations of IFN types is a very clever idea that we did not think about. Instead, we decided to start with a more simple study testing individual interferons rather than combinations. This is a reasonable idea for a future experiment. Regarding Type III IFN being dampened by types I or II, we do not believe that this was the case. The reason is that if the Type III IFN was dampened by I or II, then the type III IRG scenario would look like those elicited by Types I or II, but in fact, Type III induces its own characteristic response.

“…should clarify that the DEGs presented in panels C, D and E [Fig. 1] are strictly from the HIV-1 infected MDMs and are not the DEGs in common with the interferon treatments”. Response: Thank you for asking us to do this very important clarification. The DEGs in panels C, D and E include all the differentially regulated genes upon HIV infection regardless of their possible role in IFN responses.

“The authors should quantify that the HIV-1 infection condition is “most like Type III”, as they state in the text.”  Response: Thank you for this very appropriate criticism. As the reviewer states, we did not provide a quantitative measure of ‘proximity’ between conditions. We have now resolved this by calculating the Euclidean Distance and representing it in  bar graphs. This was actually done for Figures 1, 4, 5, 6, 7 and S-3.

Figure 2: (this is really Supplementary Figure S-1)

“The Western blots need to include MW markers”. Response: This has been done (Figure S1)

“The Nef blot has a massive water bubble and should be repeated”. Response: We have repeated all the western blots (figure S1).

“The authors state that they couldn’t blot for Vpr as no antibody exists for it. Blots for Vpr have been reported in the literature”. Response: This has been done. We used an antibody obtained from Proteintech and were able to detect Vpr (Figure S1).

“The Figure legend states that the Nef blot is panel B when it’s actually C”. Response: The new Figure S1 is now correct.

“In Figure 2a and b, the authors should provide error bars, an indication of whether variance is represented as SEM or SD, and include statistical testing.” Response: The measurements shown in this figure were done in duplicate and therefore it would be inappropriate to draw SEM or SD. The goal of this experiment was to confirm that the infections by wt and mutant viruses achieved similar levels of infection. The results show that by two different measures (%mCD24+ by flow cytometry and mRNA fold-change for mCD24 from the RNAseq) the infections were similar across the board, which is an essential quality control for the subsequent RNAseq experiments.

“…   authors should present their gating strategies. Was the respective gating performed with an FMO control?” Response: We have added further clarification to the gating strategy in the Materials and Methods section.  The FMO control was indeed done, by gating on mock-infected cells.

Figure 3 

The authors should provide a rationalization for looking at IDO1 and PLK1. We have now included this in the Results section 2.3.

“Here the authors use MDMs also infected with Δall accessory genes (DDDD) virus. It is unclear why this virus was not used in prior experiments as a functional negative control?” Response: The delta 4 virus was not available when we first studied HIV-1 infection of macrophages, and its construction was finished when many experiments had already been conducted. Rather than incorporating this new mutant virus in recent experiments but not in the earlier, foundational experiments, we have now removed the data regarding this mutant, so that the panel of viruses used from beginning to end of the study is now unchanged.

“Authors need a statistical analysis for these experiments and while they acknowledge their RNA-seq and RT-qPCR experiments for IDO1 don’t agree with each other they need to explain this.” Response: we replaced the previous Figure 3 with a newer version based on new experiments.  This new figure includes statistics, which are described in the legend, and the RT PCR results for IDO1 and PLK1 nicely agree with RNAseq results.

Figure 4 and 5: The authors acknowledge that 50% of the Vif-induced upregulated genes are actually potently downregulated by Type I and Type II interferons. Similar results were observed for Vpr. They found most of these genes were cell cycle regulators and resembles a Type III interferon response. But considering Type I and II interferons surely exist in vivo, then this may have very little significance. This should be discussed.” Response:  The reviewer is correct in asserting that HIV infection induces a collection of DEG that overlaps with DEG observed with each of the three types of IFN. The endpoints we sought to pursue in our study were (a) to dissect the individual contributions of accessory genes in inducing such a response; and (b) to ascertain the similarity or overlap between the HIV-1-induced response and those responses elicited by individual interferons. Our assertion that the HIV-1-induced response resembles type III interferon more does not exclude that there is also a considerable overlap with types I and II, as can be seen in Figure 1A. To address this comment, we calculated the Euclidean Distances in all the relevant figures (1, 4, 5, 6, 7 and S-3). We hope the reviewers will agree that Euclidean Distances provide a good quantitative estimate of the proximity between conditions.

“Figure 6: It is unclear how the protein interaction network maps were generated. Are these DEGs? It says they are in the figure legend, but doesn’t provide that information in the text.”  Response: We have added descriptions in the figure legends regarding the protein interaction network and added clarification in the corresponding lines of the Results section.

Reviewer 2 Report

In this manuscript titled “HIV-1 Accessory Proteins Impart a Modest Interferon Response and Up-Regulate Cell Cycle Related Genes in Macrophages”, the authors did an extensive transcriptome analysis on the monocyte-derived macrophages (MDMs) infected with HIV-1 or stimulated with IFN. Through comparison among HIV-1 infected MDMs versus IFN stimulated MDMs, the authors claimed that HIV-1 infection stimulated transcriptome changes are most similar to IFN-l. By using HIV-1 constructs which are deficient in Vif, Vpr, Vpu, or Nef, the authors found out the transcription changes of HIV-1 infected MDMs regulated by each of these accessory proteins. Overall, this is an excellent study on the accessory protein regulated gene expression in macrophages, which will provide novel and valuable information on how these accessory proteins regulate IFN responses. However, this manuscript is poorly organized, for example, the figure label and sequence are totally messed up, which makes me a little hard to follow the logic flow. Besides, there is a serious flaw in the comparison between HIV-1 and IFN induced transcriptome changes. I think this is a potentially significant study with additional data and a better organization of the figures.

Major concerns:

  1. The authors claimed that “HIV-1 up-regulated genes were present among genes upregulated by all four IFNs tested, with IFN-l stimulation sharing the largest number of common up-regulated genes with those up-regulated by HIV-1Bal” (line 90-92). IFN-l stimulation shares more similarities with HIV-1 infection compared to other IFN stimulation could simply be because the dosage of IFN-l used in this study happens to be the one with more physiological relevance to HIV-1 infection. It is hard to compare the transcriptome changes among different IFN with a single dosage. It would be necessary to use different dosage of each IFN to stimulate MDMs and to look at the transcripts level of several “key” genes (those genes tested or analyzed in this study). Does the different dosage of the same IFN induced largely different transcript level changes or relatively similar changes? For example, for those genes upregulated by IFN-l, will they be downregulated with a different dosage? Does IFN-l stimulated upregulation still share the biggest similarity with HIV-1 with different dosage?

  1. As mentioned above, the figure number is extremely confusing. For example, the figure “HIV-1 infection induces a type-III IFN-like response in macrophages” is labeled as figure 1; the figure “Western blot analysis of protein lysates from MDMs infected with wild-type (WT) and HIV-1Bal mutants from cell culture aliquots cultured in parallel with cultures harvested and subjected to RNA-seq” is also labeled as figure 1. I assume the latter is supplement figure 1?

  1. The Western blot in (supplement?) figure 1c certainly needs a repeat and a clearer picture.

Minor concers:

Typo needs to be corrected, for example, “IFN-g Is the sole member of the type…” (line 60) should be “IFNγ is the sole member of the type…”

Author Response

“Overall, this is an excellent study on the accessory protein regulated gene expression in macrophages, which will provide novel and valuable information on how these accessory proteins regulate IFN responses.” Response: We thank the reviewer for this positive remark.

“However, this manuscript is poorly organized, for example, the figure label and sequence are totally messed up …”  Response: We have carefully looked at figure legends and general organization and the manuscript is much more clear now.

Major concerns:

  1. " IFN-lambda stimulation shares more similarities with HIV-1 infection compared to other IFN stimulation could simply be because the dosage of IFN-lambda used in this study happens to be the one with more physiological relevance to HIV-1 infection…”  Response: We have done performed a dose (of IFNs) response for a the ISG KIF20A (kinesin family member 20). The results show that the direction of the change (downregulated by IFN alpha and gamma and not affected by lambda) is unaffected by the IFN concentration, but the magnitude of the change is, as one would expect. These data are show in the new Figure S-2.

  1. “As mentioned above, the figure number is extremely confusing.” Response: We apologize for this. We have corrected the figure numbering very carefully.

  1. “The Western blot in (supplement?) figure 1c certainly needs a repeat and a clearer picture.” Response: We have repeated all the WB. See response to Rev 1 third bullet.

Minor concerns:

“Typo needs to be corrected, for example, “IFN-g Is the sole member of the type…” (line 60) should be “IFNγ is the sole member of the type…” Response: This has been corrected.

Round 2

Reviewer 1 Report

Accept

Reviewer 2 Report

The figure legend of figure S2 and S3 is messed up again. Please read through the whole manuscript CAREFULLY when prepare the final version and make sure the figures are well-organized.

All other concerns resolved.